# Molecular Epidemiology of *tet*(A)-v1-Positive Carbapenem-Resistant *Klebsiella pneumoniae* in Pediatric Patients in a Chinese Hospital

**DOI:** 10.3390/antibiotics14090852

**Published:** 2025-08-22

**Authors:** Chen Xu, Chunli Li, Yuanyuan Li, Xiangkun Zeng, Yi Yang, Mi Zhou, Jiani Jiang, Yunbing Li, Guangfen Zhang, Xiaofan Li, Jiayi You, Yi Liu, Lili Huang, Sheng Chen, Ning Dong

**Affiliations:** 1Department of Emergency Medicine, Second Affiliated Hospital, School of Public Health, Zhejiang University School of Medicine, Hangzhou 310058, China; chen.xu@connect.polyu.hk (C.X.); chunli0620@163.com (C.L.); 20234221082@stu.suda.edu.cn (X.Z.); yi.yang@zju.edu.cn (Y.Y.); jianijiang@zju.edu.cn (J.J.); f17864190030@163.com (G.Z.); 0925058@zju.edu.cn (X.L.); youjiayi97@163.com (J.Y.); 3200100321@zju.edu.cn (Y.L.); 2Department of Laboratory Medicine, School of Medicine, Jiangsu University, Zhenjiang 212013, China; 3Liangzhu Laboratory, Zhejiang University School of Medicine, Hangzhou 311121, China; 4Department of Medical Microbiology, School of Basic Medical Sciences, Suzhou Medical College of Soochow University, Suzhou 215123, China; 5Department of Medical Microbiology, Experimental Center, Suzhou Medical College of Soochow University, Suzhou 205123, China; liyuanyuan@suda.edu.cn (Y.L.); ybli0202@suda.edu.cn (Y.L.); 6Department of Pharmacy, Children’s Hospital of Soochow University, Suzhou 215028, China; zm@suda.edu.cn; 7Laboratory Department, Children’s Hospital of Soochow University, Suzhou 215028, China; 8State Key Lab of Chemical Biology and Drug Discovery and the Department of Food Science and Nutrition, The Hong Kong Polytechnic University, Hung Hom, Hong Kong SAR, China

**Keywords:** carbapenem-resistant *Klebsiella pneumoniae*, tigecycline resistance, *tet*(A)-v1, molecular epidemiology

## Abstract

**Background:** The emergence and spread of the tigecycline resistance gene *tet*(A)-v1 in carbapenem-resistant *Klebsiella pneumoniae* (CRKP) poses significant public health challenges. However, the prevalence of *tet*(A)-v1-positive CRKP, especially in pediatric patients, remains poorly understood. This study aims to address the gap by performing an in-depth analysis of isolates collected from a children’s hospital in China. **Methods**: A 4-year retrospective study was conducted in the children’s hospital in Suzhou, China. Non-duplicated specimens were obtained from pediatric patients, and antimicrobial susceptibility profiles were assessed. Whole-genome sequencing and bioinformatics analyses were conducted to characterize the genetic background, antimicrobial resistance determinants, hypervirulence-associated genes, diversity of *tet*(A)-v1-carrying plasmids, the genetic environment of *tet*(A)-v1, and the potential for clonal transmission. Conjugative transferability of *tet*(A)-v1-carrying plasmids was also evaluated via conjugation assays. **Results**: Of the 73 *tet*(A)-v1-positive CRKP isolates from pediatric patients, 10.96% were non-susceptible to tigecycline. These isolates exhibited high genetic diversity, spanning across 13 STs (sequence types), with ST17 being predominant. Three carbapenemases were identified, with IMP being the most common. Isolates from diverse backgrounds, such as ST17, ST20, ST323, ST792, and ST3157, demonstrated evidence of clonal transmission. The *tet*(A)-v1 gene was located on 14 distinct plasmids across seven replicon types, with IncFIA/IncHI1 and IncFII being most commonly detected. All *tet*(A)-v1-carrying plasmids were multidrug-resistant, and 68.49% were conjugatively transferable, indicating a high potential for horizontal transfer. Four genetic contexts bordering *tet*(A)-v1 were identified, which points to active clonal dissemination. **Conclusions**: Although limited to a single hospital, this study represents one of the first in-depth investigations of *tet*(A)-v1-positive CRKP in pediatric patients, providing valuable insights into the prevalence and spread of *tet*(A)-v1 in this vulnerable group. These findings emphasize the urgent need for enhanced surveillance and infection control measures to curb the spread of *tet*(A)-v1-positive CRKP in pediatric healthcare environments, offering critical insights to mitigate its public health impact.

## 1. Background

Carbapenem resistance has become increasingly prevalent in Enterobacterales, particularly in *Klebsiella pneumoniae*. Previous studies indicated that carbapenem-resistant *K. pneumoniae* (CRKP) isolates accounted for 64% of carbapenem-resistant Enterobacteriaceae (CRE) infections in China [1,2]. Infections caused by CRKP are associated with high mortality rates and could lead to a variety of serious conditions, such as pneumonia, liver abscesses, urinary tract infections, and bloodstream infections [3]. CRKP strains usually possess a variety of antimicrobial resistance genes, which restrict available treatment options [4].

Tigecycline is one of the ‘last-resort’ antibiotics for treating infections caused by Gram-negative bacteria, including CRKP. It inhibits protein translation by reversibly binding to the A site of the 30S subunit of the bacterial ribosome [5]. However, its extensive use has been accompanied by a notable increase in tigecycline-resistant strains [6]. Tigecycline resistance is known to be mediated by diverse mechanisms, including the overexpression of chromosomal efflux pumps belonging to the resistance–nodulation–division (RND) family, such as OqxAB and AcrAB, mutations in the ribosomal binding site, notably in the *rpsJ* gene, and the expression of plasmid-mediated mobile tigecycline-resistance genes [7]. Among these mechanisms, plasmid-mediated resistance is particularly concerning due to its potential in facilitating horizontal gene transfer among different hosts [8]. To date, several plasmid-mediated tigecycline resistance genes have been reported, including *tet*(X) variants that encode tigecycline-inactivating enzymes, the *tmexCD1-toprJ1* gene cluster and its variants encoding the RND efflux pump, as well as some *tet*(A) variants that encode the major facilitator superfamily (MFS) efflux pump [9]. Studies in China have reported that the *tmexCD*-*toprJ* and *tet*(X) genes are sporadically distributed in clinical isolates of *K. pneumoniae*, with carriage rates for both genes remaining below 1% [10,11]. In comparison, a previous study demonstrated that 75.8% of ST (Sequence Types) 11 CRKP strains carrying *tet*(A) variants exhibited non-susceptibility to tigecycline [12].

*K. pneumoniae* strains harboring the wild-type *tet*(A) gene generally remain susceptible to tigecycline. However, mutations in *tet*(A) may result in alterations in the transmembrane region of the MFS efflux pump, potentially conferring tigecycline resistance to the host strain [13]. To date, three *tet*(A) variants associated with tigecycline resistance have been reported, including type I (*tet*(A)-v1), type II (*tet*(A)-v2), and type III (*tet*(A)-v3), with type I being dominant. Tet(A)-v1 contains seven amino acid substitutions (I5R, V55M, I75V, T84A, S201A, F202S, and V203F). The *tet*(A)-v1 variant has been shown to increase the minimum inhibitory concentration (MIC) of tigecycline by four- to eight-fold in *K. pneumoniae* [14].

The prevalence of *tet*(A)-v1 in CRKP has been rarely documented, especially in the pediatric population [12,15,16,17]. To fill this gap, we conducted a comprehensive investigation into the characteristics of *tet*(A)-v1-positive CRKP among pediatric patients in China. To the best of our knowledge, this is the first molecular epidemiological study focusing on *tet*(A)-v1-positive CRKP in pediatric patients. The findings not only enhance our understanding of the molecular mechanisms underlying *tet*(A)-v1-positive CRKP infections, but also offer valuable insights for guiding targeted interventions against the growing threat of antibiotic resistance in this vulnerable population.

## 2. Results

### 2.1. Antimicrobial Resistance Profiles of tet(A)-v1-Positive CRKP

A total of 73 *tet*(A)-v1-positive CRKP isolates were collected from pediatric patients. All these isolates were resistant to at least three classes of antibiotics. Apart from one isolate that was susceptible to ampicillin, all isolates showed intrinsic resistance to ampicillin and exhibited resistance to tetracycline. A total of eight isolates (10.96%) were non-susceptible to tigecycline, including six isolates (8.22%) resistant to tigecycline (MIC ≥ 8 μg/mL). The resistance rates to β-lactam antibiotics ceftazidime, ceftriaxone, ampicillin/sulbactam, cefotetan, imipenem, cefepime, piperacillin/tazobactam, and aztreonam were 100%, 97.26%, 97.26%, 95.89%, 90.41%, 79.45%, 52.04%, and 21.92%, respectively, and those for the quinolone antibiotics ciprofloxacin and levofloxacin were 72.60% and 32.88%, respectively (Appendix A). The resistance rate of these *tet*(A)-v1-positive CRKPs to trimethoprim/sulfamethoxazole was 73.97%, and those for the aminoglycoside antibiotics amikacin, gentamicin, and tobramycin were 12.33%, 12.33%, and 26.03%, respectively. Among the tested antibiotics, the lowest resistance rate was observed for polymyxin B (Appendix A).

### 2.2. Genetic Background of tet(A)-v1-Positive CRKP

Phylogenetic analysis suggested that these *tet*(A)-v1-positive CRKP were genetically diverse, which belonged to 13 different sequence types, including ST17 (*n* = 28), ST3157 (*n* = 17), ST792 (*n* = 14), ST20 (*n* = 3), ST323 (*n* = 2), ST1306 (*n* = 2), ST15 (*n* = 1), ST54 (*n* = 1), ST307 (*n* = 1), ST4424 (*n* = 1), ST11 (*n* = 1), ST1662 (*n* = 1), and ST2128-1LV (*n* = 1). PopPUNK clustering further assigned these isolates into 12 sequence clustering (SC) types, including SC11_15_16_108_183_207_210_243_277 (*n* = 31), SC297 (*n* = 17), SC254 (*n* = 14), SC298 (*n* = 2), SC20 (*n* = 2), SC1 (*n* = 1), SC2 (*n* = 1), SC8 (*n* = 1), SC299 (*n* = 1), SC300 (*n* = 1), SC301 (*n* = 1), and SC302 (*n* = 1). Apart from ST17 and ST20, which both belonged to SC11_15_16_108_183_207_210_243_277, each ST type belonged to a different SC type. The 73 *tet*(A)-v1-positive CRKP isolates belonged to 11 distinct serotypes, with the dominant types being KL25 and KL30, which accounted for 57.53% (*n* = 42) and 23.29% (*n* = 17), respectively. Notably, all ST17 and ST792 isolates belonged to KL25, and all ST3157 belonged to KL30 (Figure 1).

### 2.3. Antimicrobial Resistance Genes in tet(A)-v1-Positive CRKP

All *tet*(A)-v1-positive CRKP isolates carried multiple antimicrobial resistance genes, with the number ranging from 8–22 (Figure 2 and Appendix A). Most isolates (70/73, 95.89%) harbored no less than 10 resistance genes, which were in line with their multidrug resistance phenotypes. Of the 73 isolates, 43.84% (*n* = 32) carried ESBL genes, including 27.40% (*n* = 20) harboring *bla*_CTX-M-14_, 12.33% (*n* = 9) carrying *bla*_CTX-M-3_, 2.74% (*n* = 2) harboring *bla*_CTX-M-15_, and 1.37% (*n* = 1) harboring *bla*_CTX-M-65_. Each *tet*(A)-v1-positive CRKP isolate carried one carbapenemase gene, which contributed to the carbapenem resistance phenotype. Four different types of carbapenemase genes were detected among these isolates, with *bla*_IMP-4_ being dominant (*n* = 37, 50.68%), followed by *bla*_NDM-5_ (*n* = 29, 39.73%), *bla*_NDM-1_ (*n* = 6, 8.22%), and *bla*_KPC-2_ (*n* = 1, 1.37%). ST17 isolates were predominantly associated with *bla*_IMP-4_ (19/28, 67.88%), all ST3157 isolates carried *bla*_IMP-4_ and all ST792 isolates carried *bla*_NDM-5_ (Figure 1 and Appendix A).

### 2.4. Hypervirulence-Associated Genes in tet(A)-v1-Positive CRKP

The distribution of hypervirulence-associated genes, including those encoding yersiniabactin, colibactin, aerobactin, salmochelin, RmpADC, and RmpA2, in the *tet*(A)-v1-positive CRKP isolates were identified. Hypervirulence genes were sporadically distributed in these *tet*(A)-v1-positive CRKP isolates. Among the 73 isolates, 47.95% encoded yersiniabactin, including isolates belonging to ST17 (*n* = 19), ST792 (*n* = 14), ST54 (*n* = 1), and ST11 (*n* = 1) (Figure 1, Appendix A). Furthermore, all ST792 isolates carried both yersiniabactin and colibactin, while none of the other strains carried colibactin. Aerobactin was absent in all *tet*(A)-v1-positive CRKP isolates, except those belonging to ST11 (*n* = 1) and ST2128-1LV (*n* = 1). Salmochelin was found to be absent in all 73 isolates. Notably, only the one ST11 isolate encoded four virulence factors, including yersiniabactin, aerobactin, RmpADC, and RmpA2. In contrast, the other strains lack factors such as RmpADC and RmpA2 (Figure 2).

### 2.5. Clonal Transmission of tet(A)-v1-Positive CRKP

To identify potential clonal isolates, the SNP distances between isolates of different STs were analyzed, with SNP ≤ 25 defined as clonal relatedness [18] (Figure 3, Appendix A). Based on the SNP similarity matrix across genomes, a threshold of 25 SNPs was selected to generate the SNP heatmap. Pairwise SNP analysis indicated that certain *tet*(A)-v1-positive CRKP isolates had undergone clonal spread within the same ST (Figure 3). Clonal dissemination was observed in strains belonging to ST17, ST20, ST323, ST3157, and ST792. Specifically, more than one clone was observed among isolates belonging to ST17/KL25 (cluster I (*n* = 9), cluster II (*n* = 19)) (Appendix A), ST1306/KL146 (cluster I (*n* = 1), cluster II (*n* = 1)) (Appendix A), and ST3157/KL30 (cluster I (*n* = 3), cluster II (*n* = 13), cluster III (*n* = 1)) (Appendix A). All isolates were from different patients, suggesting the clonal transmission of *tet*(A)-v1-positive CRKP isolates of diverse genetic backgrounds, likely due to hospital outbreaks that facilitated their spread.

### 2.6. Diversity of tet(A)-v1-Carrying Plasmids

Genetic analysis suggested the *tet*(A)-v1 genes in all the 73 CRKP isolates were located on plasmids. The plasmid genetic content analysis by BLAST search in the NCBI database suggested that the *tet*(A)-v1 genes in these isolates were located on plasmids of 14 unique types ranging from 51 Kb to 315 Kb, which belonged to 7 different replicon types including IncFIA/IncHI1 (*n* = 30), IncFII (*n* = 21), IncA/C2 (*n* = 9), IncFIB(κ) (*n* = 5), IncR/n (*n* = 4), IncFIB (*n* = 3), and IncHI2 (*n* = 1) (Figure 1, Figure 2, Figure 4 and Appendix A). *tet*(A)-v1 genes in isolates with different genetic backgrounds were carried by different types of plasmids. Among these, the pMTY13754_IncF-like plasmid was the most predominant. pMTY13754_IncF is an IncFIA/IncHI1 plasmid, with the length being 131,450 bp. It can align with 30 isolates from this study, including those belonging to ST3157/KL30 (*n* = 16) and ST792/KL25 (*n* = 14) (Figure 4 and Appendix A). pMTY13754_IncF possesses several antibiotic-resistance genes, including *tet*(A)-v1, *sul2*, *floR*, *dfrA14*, *qnrS1*, and *bla*_LAP_-like. Apart from the 16 ST3157 isolates carrying the pMTY13754_IncF-like plasmid, one ST3157 isolate (BSIKP_68) carried the IncFII K15_unnamed-like *tet*(A)-v1-carrying plasmid (Appendix A). Two types of *tet*(A)-v1-carrying plasmids were observed in ST17 isolates, which were p205880_ct1/2-like (153,373 bp, IncA/C2, *n* = 9) and pGSU10-3-2-like (134,879 bp, IncFII, *n* = 19) (Appendix A and Appendix A). Detailed information on the *tet*(A)-v1-carrying plasmids is provided in the Appendix A.

### 2.7. Conjugative Transferability of tet(A)-v1-Carrying Plasmids

A conjugation assay was performed to test the transferability of the *tet*(A)-v1-carrying plasmids. Among the 73 isolates, 68.49% (*n* = 50) successfully transferred the *tet*(A)-v1-carrying plasmid to *E. coli*. The highest conjugation frequencies were observed in ST3157 isolates, ranging from 10^−3^ to 10^−6^. The conjugation frequencies of *tet*(A)-v1-carrying plasmids in other STs generally ranged from 10^−7^ to 10^−8^. However, despite our efforts, the *tet*(A)-v1-carrying plasmids from certain STs, including ST15, ST1662, ST20, ST307, and ST54, could not be conjugatively transferred to the *E. coli* recipient (Appendix A). The widespread conjugative transferability of *tet*(A)-v1-positive plasmids in CRKP highlights the potential for broad dissemination of the *tet*(A)-v1 gene through horizontal transfer.

The analysis of the relationship between plasmid types and conjugative transfer showed that plasmids belonging to IncA/C2 (9/9, 100%), IncFIA/IncHI1 (24/30, 80%), IncFIB(κ) (3/5, 60%), IncFII (12/21, 57.14%), and IncR/N (2/4, 50%) were capable of horizontal transfer via conjugation (Appendix A). In particular, IncFIA/IncHI1 achieved a transfer frequency of up to 10^−3^. In contrast, plasmids of the IncFIB and IncHI2 types were not conjugatively transferable.

### 2.8. Genetic Contexts of tet(A)-v1 in CRKP

A total of four types of genetic contexts bordering *tet*(A)-v1 were identified, including type I (IS*Vsa3*-*virD*2-*floR*-*lysR*-*tet*(A)-v1-*tetR*-*hp*, *n* = 9), type II (Tn*As1*-*DMT*-*tet*(A)-v1-*tetR*-*rep*-IS*26*, *n* = 33), type III (Tn*As1*-*hp*-*DMT*-*tet*(A)-v1-*tet*R-*hel*, *n* = 9), and type IV (*hp*-*DMT*-*tet*(A)-v1-*tetR*-*rep*, *n* = 22) (Figure 5). The *tet*(A)-v1 genes were frequently associated with mobile elements such as Tn*As1*, IS*26*, and IS*Vsa3*, suggesting that they could be acquired by horizontal gene transfer. Genetic context type I was closely associated with IncA/C2 plasmids (9/9, 100%). Other types of genetic contexts were carried by diverse plasmids. Type II was carried by IncFIA/IncHI1 (30/30, 100%), IncFIB(κ) (2/5, 40%), and IncFII (1/21, 4.76%) plasmids, type III was carried by IncFIB (3/3, 100%), IncHI2 (1/1, 100%), IncR/N (3/4, 75%), and IncFIB(κ) (2/5, 40%) plasmids, and type IV was harbored by IncFII (20/21, 95.24%), IncR/N (1/4, 25%), and IncFIB(κ) (1/5, 20%) plasmids (Figure 6). These findings further underscore the complexity of the genetic context harboring the *tet*(A)-v1 gene.

## 3. Discussion

CRKP poses a significant global health challenge. The emergence of resistance to last-line antibiotics, tigecycline and colistin, has further complicated clinical management and treatment options. The role of the *tet*(A)-v1 gene in mediating transferable tigecycline resistance in CRKP has gained importance since it was first reported in 2017 [19]. Despite its clinical importance, the prevalence of these *tet*(A)-v1 variants in the clinical setting remains largely uncharacterized [20,21].

In this study, we investigated the molecular epidemiology of 73 *tet*(A)-v1-positive CRKP isolates collected from a children’s hospital in China. A proportion of 10.96% of these isolates were non-susceptible to tigecycline, and all isolates exhibited multidrug resistance, which posed a significant challenge to clinical treatment. The tigecycline MICs of the *tet*(A)-v1-positive isolates ranged from 0.25 to 8 μg/mL, with the majority showing low MICs, consistent with previous findings that *tet*(A)-v1 confers only low-level tigecycline resistance [14]. It has been reported that mutations in *tet*(A)-v1, an MFS family efflux pump, may increase intracellular accumulation of tigecycline as a substrate, potentially leading to tigecycline resistance [19,22]. Although *tet*(A)-v1 conferred only low-level tigecycline resistance in our study, its widespread prevalence in the hospital poses significant therapeutic challenges.

Our study showed that there are significant differences between *tet*(A)-v1-positive CRKP isolates obtained from pediatric patients and those from adult patients. Most *tet*(A)-v1-positive CRKP isolates reported in adults belonged to ST11 [12,15], but those in this pediatric cohort were ST17, ST3157, and ST792. Clonal transmission of ST17 and ST3157 isolates was detected, which could be associated with the prevalence of these STs in pediatrics. Moreover, *tet*(A)-v1 genes were conjugative in all the three ST types. Furthermore, KPC-2 is the predominant carbapenemase in CRKP in adult patients [23,24], while in *tet*(A)-v1-positive CRKP isolates from children, IMP was dominant, followed by NDM, and KPC. This observation highlights the distinct differences in CRKP profiles between pediatric and adult populations, emphasizing the necessity for enhanced surveillance of CRKP in children.

*tet*(A)-v1-carrying plasmids in CRKP isolates from pediatric patients were highly diverse. In addition to the previously reported IncFII and IncFIB(κ) plasmids [12,16], five additional Inc types were identified in this study, including IncR/N, IncA/C2, IncFIB, IncHI2, and IncFIA/IncHI1. The conjugative transferability of some plasmids contributed significantly to the dissemination of *tet*(A)-v1 in CRKP. In particular, IncFIA/IncHI1 showed the highest conjugation frequency of up to 10^−3^, which could be associated with its high prevalence. In addition to plasmid-mediated horizontal transmission, clonal spread of *tet*(A)-v1-positive CRKP was detected, both of which contributed to the widespread dissemination of *tet*(A) in the pediatric population [12,15,16]. Clonal transmission was particularly pronounced in isolates belonging to ST17, ST3157, and ST792. This dissemination has the potential to exacerbate treatment challenges and complicate infection management.

The identification of genetic environments is essential for elucidating the dissemination of antibiotic resistance genes. In this study, a total of four different genetic contexts of *tet*(A)-v1 were identified, among which types II and III were highly identical to those reported previously [12,15,16,17]. The IncFII and IncFIB *tet*(A)-carrying plasmids identified in this study were also commonly found in adults, as reported previously. Moreover, the genetic contexts of the *tet*(A) genes in these plasmids from adult isolates were highly similar to the type II and type III genetic contexts of *tet*(A) in this study [12,15,16,17].

Core-pan genome analysis suggested that *tet*(A)-v1-positive CRKP isolates have a relatively large pan-genome, which could constantly acquire genetic material and evolve into novel superbugs. In addition, some *tet*(A)-v1-CRKP isolates carried genes associated with hypervirulence, suggesting the emergence of superbugs that are simultaneously resistant to last-line antibiotics (tigecycline and carbapenems), hypervirulent, and possibly highly transmissible, which could pose a significant clinical challenge.

We acknowledge several limitations in this study. First, this study was conducted in a single children’s hospital, which may limit the generalizability of the findings to other geographic regions or adult populations. Second, the sample size of *tet*(A)-v1-positive CRKP isolates (*n* = 73) was relatively small, and larger-scale surveillance would provide more robust epidemiological insights. Third, this study focused primarily on genomic and phenotypic characterization, whereas in vivo experiments or longitudinal tracking of transmission dynamics were not performed to fully assess the clinical impact and persistence of these strains. Future research should focus on establishing nationwide multicenter cohort studies across age groups, integrating functional genomics and phenomics platforms, developing molecular epidemiology-based surveillance networks, and creating resistance mechanism-targeted control strategies.

In conclusion, this is the first study to comprehensively investigate the molecular epidemiology of *tet*(A)-positive CRKP in pediatric patients. Our results suggested that ST17 is most prevalent among pediatric isolates. Seven types of *tet*(A)-carrying plasmids and four genetic contexts of *tet*(A)-v1 were identified. Vertical transmission of *tet*(A)-v1-positive CRKP was observed in five STs. *tet*(A)-v1 in 68.49% of the isolates was transferable, and hypervirulence genes were also detected in some isolates. These findings highlight the high diversity as well as the vertical and horizontal transferability of *tet*(A)-v1-positive CRKP in pediatric patients. This study underscores the complexity of CRKP transmission dynamics in pediatric populations, and also provides critical insights for the development of targeted interventions.

## 4. Methods

### 4.1. Study Design and Setting

This retrospective study analyzed data from 1 September 2016, to 31 October 2020, sourced from the microbiology laboratory at the Children’s Hospital of Soochow University (CHSU, Suzhou, China). As a tertiary pediatric medical center and the largest children’s hospital in Suzhou, CHSU exclusively serves pediatric patients and has over 1300 dedicated pediatric beds. It is the primary referral hospital for pediatric care in Suzhou and the surrounding areas, covering a significant portion of the local pediatric population. The CHSU microbiological laboratory processes samples from all hospital departments and external health facilities throughout its catchment areas. On average, the laboratory handles 4000 culture samples monthly, performing antimicrobial susceptibility testing as clinically required.

### 4.2. Bacteria Isolates and Clinical Data Collection

A total of 140 non-duplicate CRKP isolates were isolated from blood, urine, sputum, and fecal specimens of 140 pediatric patients aged from 1 day to 14 years from CHSU, during the period between September 2016 and October 2020. The isolates were subjected to species identification using MALDI-TOF MS (Bruker Daltonik GmbH, Bremen, Germany) and 16S rRNA gene sequencing [25]. The presence of *tet*(A)-v1 was screened by PCR and Sanger sequencing using primers reported previously, and further validated by analyzing the whole-genome sequences (see below) [19]. The genomic information of all *tet*(A)-v1-positive CRKP isolates were presented in Appendix A.

### 4.3. Antimicrobial Susceptibility Testing

Antimicrobial susceptibility testing was performed by broth microdilution to determine the MIC. The antibiotics tested were listed in Appendix A. The results for all antibiotics except tigecycline were interpreted according to the CLSI guideline [26]. Tigecycline susceptibility was interpreted in accordance with the FDA identified interpretive criteria [27]. *E. coli* ATCC 25922 was used as the quality control.

### 4.4. Conjugation Assay

Conjugation was performed to evaluate the transferability of the *tet*(A)-v1 in CRKP isolates [28]. Rifampicin-resistant *E. coli* EC600 or sodium azide-resistant *E. coli* J53 was used as a recipient (both sensitive to oxytetracycline), and *tet*(A)-v1-carrying isolates in this study were used as donors. The recipients and donors were mixed with a 1:1 ratio and cultured at 37 °C for 16 h. Transconjugants were screened on LB agar plates containing 4 μg/mL oxytetracycline and 600 μg/mL rifampicin or 150 μg/mL sodium azide. Successful transconjugants were validated by antimicrobial susceptibility testing and PCR targeting the *tet*(A)-v1 gene.

### 4.5. DNA Extraction and Whole-Genome Sequencing

Bacterial genomic DNA was extracted from overnight cultures using the genomic DNA extraction kit (Vazyme, Nanjing, China) according to the manufacturer’s instructions. The purity and concentration of the extracted DNA was measured using a NanoDrop spectrophotometer (Thermo Scientific, Waltham, MA, USA). Genomic DNA was sequenced by short-read sequencing (2 × 150 bp) on the Illumina HiSeq 2500 platform [29]. Since the genetic structure of the *tet*(A)-v1-carrying plasmid or the genetic context in some isolates was difficult to obtain with second-generation short-read sequencing, four representative isolates (BSIKP-28, terRKP-23, terRKP-228, and GJY-G6) were subjected to long-read sequencing using the MinION platform (Oxford Nanopore Technologies, Oxford, UK) [29].

### 4.6. Bioinformatics Analysis

De novo genome assembly was performed with SPAdes 3.15.1 using the short sequencing reads [30]. Hybrid assembly of both short and long sequencing reads was conducted using Unicycler v0.5.0 [31]. The assembled genome sequences were annotated using the RAST tool and edited manually [32]. Genotyping was performed using Kleborate, which included species identification, multilocus sequence typing (MLST), serotyping, and the identification of antimicrobial resistance genes and virulence genes [33]. The assembled genomes were clustered using PopPUNK v2.7.0 for sequence cluster (SC) analysis [34]. Plasmid replicons and insertion sequences (ISs) were analyzed using PlasmidFinder 2.1 and ISfinder, respectively [35,36]. The locations of *tet*(A)-v1 were analyzed by mapping the *tet*(A)-v1-carrying contigs to the reference plasmid/chromosome sequences. Plasmids were visualized using BLAST Ring Image Generator (v2.2.28) [37,38]. BSIKP_111, BSIKP_70, CRESXJ35, BSIKP_28, terRKP_612, and BSIKP_67 were randomly selected as the reference genomes representing ST17, ST792, ST3157, ST20, ST323, and ST1306, respectively. Single-nucleotide polymorphisms (SNPs) were identified by mapping the Illumina raw reads to the reference genome. Pairwise SNPs of isolates belonging to the same STs were calculated with snp-dists [39]. R was used to visualize this SNP difference matrix and create an SNP heatmap [40]. The heatmap of resistance genes, virulence genes, plasmid types, and genetic environments of different STs were visualized using TBtools-II v2.056 [41].

### 4.7. Phylogenetic Analysis

The phylogenetic tree of all *tet*(A)-v1-positive isolates was constructed using the Harvest Suite and then visualized and edited with Evolview-v2 [42,43]. Additional information was added to the tree, including STs, SC types, K locus types, carbapenemase genes, genetic contexts, and plasmid types. A minimum spanning tree showing the distribution of carbapenem-resistant genes was constructed based on the MLST of the isolates using GrapeTree (https://achtman-lab.github.io/GrapeTree/MSTree_holder.html) [44]. A Sankey diagram showing the relationships between different STs, plasmid replicons, and genetic backgrounds was constructed using R 4.4.1 [45].

## Figures and Tables

**Figure 1 antibiotics-14-00852-f001:**
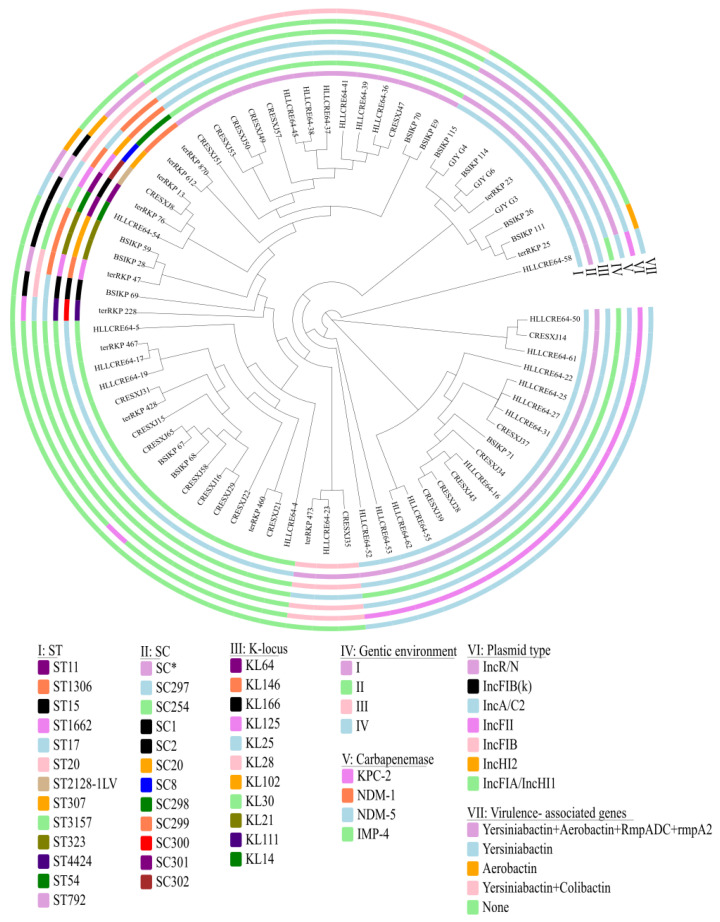
Phylogenetic tree of *tet*(A)-v1-positive CRKP. The tree was built based on the core SNPs of 73 *tet*(A)-v1-positive CRKP isolates from pediatric patients. Rings from innermost to outermost indicate the ST typs, SC types, K locus types, genetic environments of *tet*(A)-v1, carabapenemase, replicons of *tet*(A)-carrying plasmids, and hypervirulence-associated factors in these *tet*(A)-v1-positive isolates, respectively. SC* represents SC11_15_16_108_183_207_210_243_277.

**Figure 2 antibiotics-14-00852-f002:**
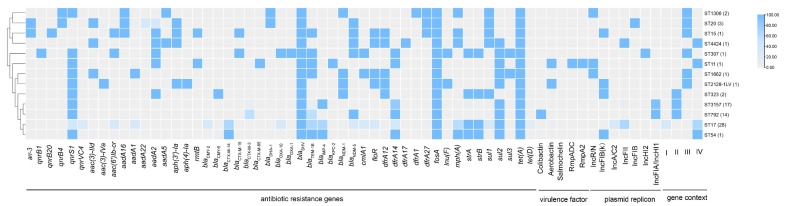
Heatmap showing the distribution of antibiotic resistance genes, hypervirulence factors, plasmids replicons, and genetic contexts among different ST types of *tet*(A)-v1-positive CRKP. The numbers in parentheses following each ST type indicate the number of isolates within the corresponding ST.

**Figure 3 antibiotics-14-00852-f003:**
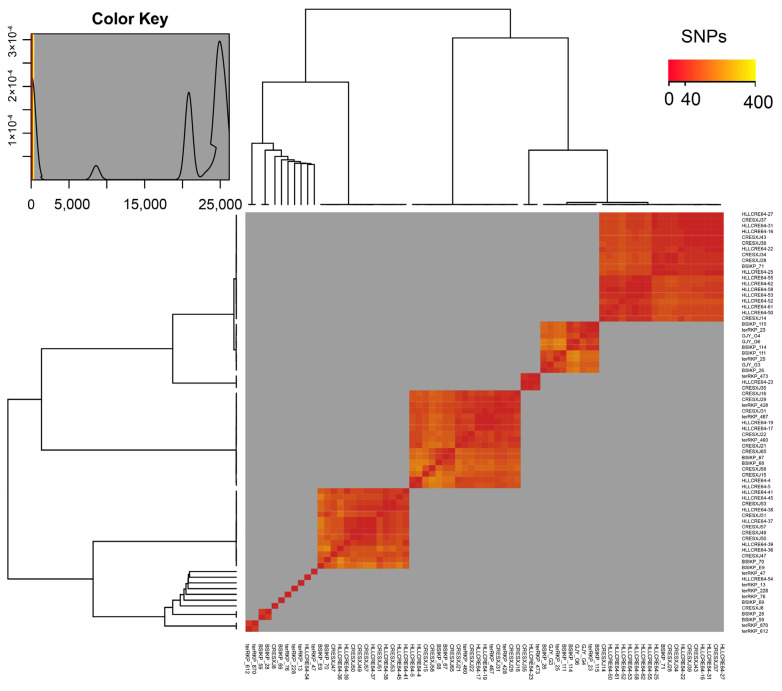
Pairwise SNP analysis of *tet*(A)-v1-positive CRKP. Heatmap of SNP differences (0–25) among *tet*(A)-v1-positive CRKP grouped by sector. Horizontal and vertical phylogenetic trees are based on the core genome sequences for each sector. The number of SNPs in the core genome between different sequenced isolates is interpreted as a distance matrix. The diagonal line represents intrasector comparisons with the three main clusters of *tet*(A)-v1-positive CRKP. The observed clustering pattern with low SNP differences (<25 SNPs) strongly suggests clonal expansion events, indicative of probable hospital outbreaks that facilitated the spread of these resistant clones.

**Figure 4 antibiotics-14-00852-f004:**
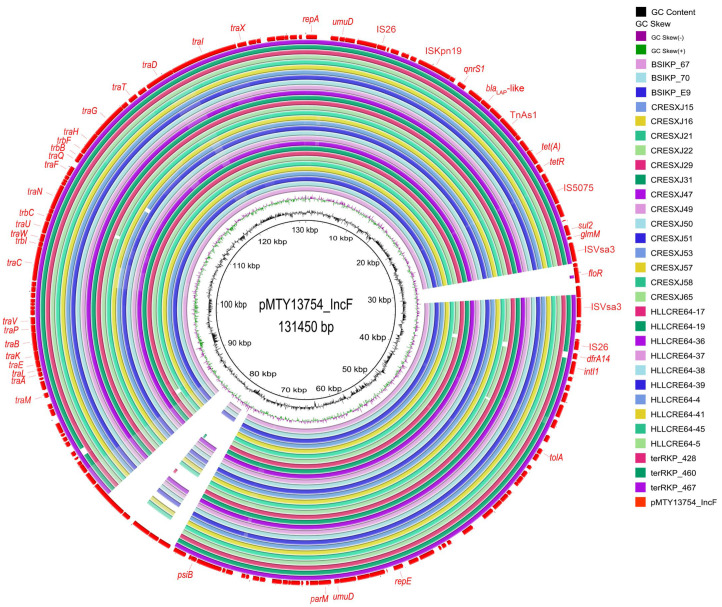
Circular alignment of pMTY13754_IncF-like plasmids. The genomes of 30 *tet*(A)-v1-carrying plasmids in this study were mapped to pMTY13754_IncF (GenBank accession: CP134363). Representative antimicrobial resistance genes and mobile genetic elements are annotated on the outermost circle.

**Figure 5 antibiotics-14-00852-f005:**
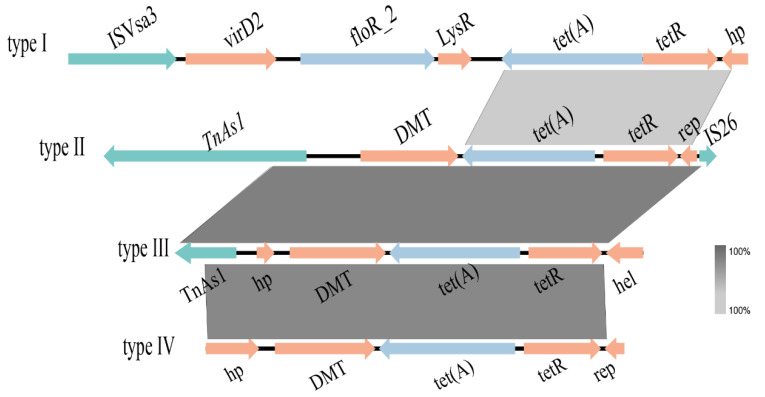
Alignment of *tet*(A)-v1 genetic contexts in CRKP. Blue, green, and orange arrows represent antimicrobial resistance genes, mobile genetic elements, and other genes, respectively.

**Figure 6 antibiotics-14-00852-f006:**
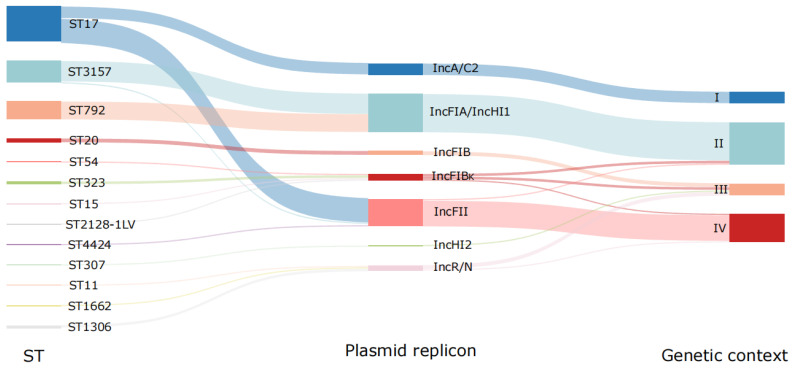
Sankey diagram showing the correlations among the ST types, plasmid replicons, and genetic contexts of *tet*(A)-v1-positive CRKP. The four genetic contexts, labeled I, II, III, and IV, represent the distinct contexts identified in this study.

## Data Availability

All data generated or analyzed during this study are included in this article and Appendix A. The assembled genome sequences of all *tet*(A)-v1-positive CRKP isolates in this study were deposited in the NBCI database under BioProject accession number PRJNA1177636.

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
