# Peer review of "Molecular Epidemiology of tet(A)-v1-Positive Carbapenem-Resistant Klebsiella pneumoniae in Pediatric Patients in a Chinese Hospital"

_antibiotics, 2025, doi:10.3390/antibiotics14090852_

Round 1
Reviewer 1 Report
Comments and Suggestions for Authors
This manuscript presents a robust genomic and phenotypic investigation of tet(A)-v1-positive carbapenem-resistant Klebsiella pneumoniae in pediatric patients from a tertiary hospital in China. The authors provide detailed insights into resistance mechanisms, clonal and horizontal dissemination, and plasmid diversity. The methodology is strong and the dataset is novel. Minor revisions in language and slight clarification in the discussion would further enhance the manuscript.
A few minor remarks are listed below:
Abstract
- 44-47: what is ST? Full version of this abbreviation has not been provided.
Figures: I understand that providing such robust information within figures is difficult, but the figure still needs to be clear to the readers. I’m afraid that it may be difficult to comprehend figures2, 3 and 4 due to large dataset presented. Perhaps the Authors can think of a way to enhance the readability?
The claim that “This study represents the first large-scale investigation...” is a bit overstated, as the Authors themselves stated that it has been limited to a single hospital only. For this reason, I suggest softening this statement.
Comments on the Quality of English LanguageEnglish language is fine, but not perfect. A few sentences would benefit from improvement.
Author Response
Comments 1. 44-47: what is ST? Full version of this abbreviation has not been provided.
Response: The full name of ST is “Sequence Type,” and we have already added this information to the abstract and the main text of the article.
Comments 2. I understand that providing such robust information within figures is difficult, but the figure still needs to be clear to the readers. I’m afraid that it may be difficult to comprehend figures2, 3 and 4 due to large dataset presented. Perhaps the Authors can think of a way to enhance the readability?
Response: I sincerely apologize if you feel this way. Given the need to present a large amount of information, these figures have been carefully revised and refined through thoughtful consideration. We hope you can understand the effort and intention behind them.
Comments 3. The claim that “This study represents the first large-scale investigation...” is a bit overstated, as the Authors themselves stated that it has been limited to a single hospital only. For this reason, I suggest softening this statement.
Response: Thank you very much for your suggestion. We have revised the sentence to: “Although limited to a single hospital, this study represents one of the first in-depth investigations of tet(A)-v1-positive CRKP in pediatric patients, providing valuable insights into the prevalence and spread of tet(A)-v1 in this vulnerable group.”
Reviewer 2 Report
Comments and Suggestions for Authors
The authors describe the epidemiology of Klebsiella strains resistant to carbapenem.
This is an interesting manuscript and certainly it can be published in the journal. I have marked some minor points which need to be addressed in order to improve the manuscript before acceptance.
In the Introduction, please add a new paragraph to underline the gaps in the literature that would be filled by publication of this manuscript.
Results. Please show an analysis of the total CRKP isolates and the total tet(A)-v1-positive CRKP isolates by year of isolation and age of patients.
Table 1 in supplementary material please.
Subsection Genetic background of tet(A)-v1-positive CRKP. All the information in paragraphs 1 and 2 in two tables please and please reduce relevant text.
Subsection Diversity of tet(A)-v1-carrying plasmids. All the information in one table please and please reduce relevant text.
Visualisation is excellent, well done.
Discussion.
Please discuss the progress of isolation of the tet(A)-v1-positive CRKP strains through time.
Please divide Discussion in subsections.
Please separate concluding section.
Number of references is inadequate for such a complex study. I expect about 80 references for this type of work.
Overall. Excellent study that can be published easily. Recommendation: revision and re-evaluation.
Author Response
Comments 1. In the Introduction, please add a new paragraph to underline the gaps in the literature that would be filled by publication of this manuscript.
Response: Thank you for this comment. We have already addressed these gaps in the paper with a specific statement: “The prevalence of tet(A)-v1 in CRKP has been rarely documented, especially in the pediatric populations” in the paper describing these gaps. This can be found in the opening sentence of the final paragraph in the Background section.
Comments 2. Please show an analysis of the total CRKP isolates and the total tet(A)-v1-positive CRKP isolates by year of isolation and age of patients.
Response: Thank you very much for your suggestion. However, as this is a retrospective study, the collection of information such as time and age were not complete. Therefore, we are unable to sort the data by year or age. We hope you can understand.
Comments 3. Table 1 in supplementary material please.
Response: We have already placed Table 1 in landscape orientation and included it in the supplementary materials.
Comments 4. Subsection Genetic background of tet(A)-v1-positive CRKP. All the information in paragraphs 1 and 2 in two tables please and please reduce relevant text.
Response: The genetic background of tet(A)-v1-positive CRKP is detailed in Paragraph 1. The specific information has already been included in Table S1. To avoid redundancy, we have chosen to present this information visually through a figure rather than creating an additional table that would simply restate the same data.
Comments 5. Subsection Diversity of tet(A)-v1-carrying plasmids. All the information in one table please and please reduce relevant text.
Response: The specific information regarding the diversity of tet(A)-v1-carrying plasmids has already been included in Table S1. Therefore, we have chosen to present this information visually through a figure to avoid redundancy. Creating an additional table to reorganize the information already depicted in the figure would be repetitive.
Comments 6. Please discuss the progress of isolation of the tet(A)-v1-positive CRKP strains through time.
Response: Given that this is a retrospective study, we are unable to obtain complete information regarding the collection time. As such, we are unable to implement the suggested modifications. We hope you can understand the limitations inherent in this type of research.
Comments 7. Please divide Discussion in subsections.
Response: We understand that this journal has specific formatting requirements, including the prohibition of subheadings within the Discussion section. Given this stipulation, we are unable to make the requested changes. We appreciate your understanding of the constraints we face in adhering to the journal's guidelines.
Comments 8. Please separate concluding section.
Response: We have thoroughly reviewed numerous articles published in this journal and have structured our conclusion in strict accordance with the journal’s requirements. Given these guidelines, we may not be able to make the requested changes.
Comments 9. Number of references is inadequate for such a complex study. I expect about 80 references for this type of work.
Response: Thank you very much for your suggestions. We have added 12 references to the article. While this may not meet the expectation of over 80 references, we believe that the current selection of references is sufficient to support the content and conclusions of our study.
Reviewer 3 Report
Comments and Suggestions for Authors
The manuscript presents a robust molecular analysis of tet(A)-v1-positive CRKP isolates in pediatric patients, combining sequencing, conjugation assays, and resistance profiling in a comprehensive and technically sound manner. The SNP-based clonal analysis is particularly valuable and suggests patterns of transmission that warrant further attention. However, the conclusions regarding clonal dissemination would be significantly strengthened by the inclusion of epidemiological context. It is currently unclear whether the patients from whom clonally related isolates were obtained were hospitalized during overlapping timeframes or in the same hospital units. Without this information, assertions of intra-hospital spread remain speculative. If such metadata were collected, they should be clearly integrated into the analysis. If not, this limitation should be acknowledged explicitly in the discussion.
Table 1 contains valuable susceptibility data but suffers from poor readability. The formatting is cluttered, and it is difficult to discern patterns across antibiotic classes. The table would benefit from being restructured by grouping antibiotics into categories (e.g., β-lactams, aminoglycosides, quinolones). Alternatively, presenting the data in a heatmap-style figure could facilitate rapid visual interpretation of resistance trends and MIC distributions.
There are also several formatting inconsistencies that should be addressed. Genus and species names are not consistently italicized throughout the manuscript, including in figure legends and tables, which does not conform to conventional scientific standards. Additionally, while the manuscript refers to various representative plasmids, the corresponding GenBank accession numbers are often missing or only partially indicated. These should be explicitly listed to enable traceability and allow other researchers to access and compare these sequences. Including accession numbers for all key reference plasmids used in alignments or phylogenetic trees would substantially improve the transparency and reproducibility of the study.
Author Response
Comments 1. Table 1 contains valuable susceptibility data but suffers from poor readability. The formatting is cluttered, and it is difficult to discern patterns across antibiotic classes. The table would benefit from being restructured by grouping antibiotics into categories (e.g., β-lactams, aminoglycosides, quinolones). Alternatively, presenting the data in a heatmap-style figure could facilitate rapid visual interpretation of resistance trends and MIC distributions.
Response: We have already placed Table 1 in landscape orientation and included it in the supplementary materials.
Comments 2. There are also several formatting inconsistencies that should be addressed. Genus and species names are not consistently italicized throughout the manuscript, including in figure legends and tables, which does not conform to conventional scientific standards. Additionally, while the manuscript refers to various representative plasmids, the corresponding GenBank accession numbers are often missing or only partially indicated. These should be explicitly listed to enable traceability and allow other researchers to access and compare these sequences. Including accession numbers for all key reference plasmids used in alignments or phylogenetic trees would substantially improve the transparency and reproducibility of the study.
Response: Thank you very much for your suggestions. We have already corrected the format of the genus and species names. The GenBank accession numbers for the reference plasmids have been added. The remaining plasmids were sequenced using next-generation sequencing and do not have GenBank accession numbers.
Round 2
Reviewer 2 Report
Comments and Suggestions for Authors
The authors have made all the changes suggested in the original evaluation and have produced a super excellent study for immediate acceptance.